# Spinal Meningiomas: Influence of Cord Compression and Radiological Features on Preoperative Functional Status and Outcome

**DOI:** 10.3390/cancers13164183

**Published:** 2021-08-20

**Authors:** Valentina Baro, Alessandro Moiraghi, Valentina Carlucci, Luca Paun, Mariagiulia Anglani, Mario Ermani, Andrea Saladino, Franco Chioffi, Domenico d’Avella, Andrea Landi, Andrea Bartoli, Francesco DiMeco, Karl Schaller, Luca Denaro, Enrico Tessitore

**Affiliations:** 1Academic Neurosurgery, Department of Neurosciences, University of Padova, 35128 Padova, Italy; valentina.baro@unipd.it (V.B.); domenico.davella@unipd.it (D.d.); andrea.landi@unipd.it (A.L.); luca.denaro@unipd.it (L.D.); 2Department of Neurosurgery, GHU Paris-Sainte-Anne Hospital, 75014 Paris, France; 3Division of Neurosurgery, Department of Clinical Neurosciences, Geneva University Hospitals and Faculty of Medicine, University of Geneva, 1211 Geneva, Switzerland; luca.paun@hcuge.ch (L.P.); andrea.bartoli@hcuge.ch (A.B.); karl.schaller@hcuge.ch (K.S.); enrico.tessitore@hcuge.ch (E.T.); 4Institut de Psychiatrie et Neurosciences de Paris (IPNP), UMR S1266, INSERM, IMA-BRAIN, 75014 Paris, France; 5Division of Neurology, Department of Neurosciences, University of Padova, 35128 Padova, Italy; valentina.carlucci@aopd.veneto.it; 6Neuroradiology Unit, Azienda Ospedaliera Università di Padova, 35128 Padova, Italy; mariagiulia.anglani@aopd.veneto.it; 7Department of Neurosciences, University of Padova, 35128 Padova, Italy; mario.ermani@unipd.it; 8Neurosurgery Department, Fondazione IRCCS Istituto Neurologico Nazionale “C. Besta”, 20133 Milan, Italy; andrea.saladino@istituto-besta.it (A.S.); fdimeco1@jhmi.edu (F.D.); 9Division of Neurosurgery, Azienda Ospedaliera Università di Padova, 35128 Padova, Italy; franco.chioffi@aopd.veneto.it; 10Department of Pathophysiology and Transplantation, University of Milan, 20122 Milan, Italy; 11Department of Neurological Surgery, Johns Hopkins Medical School, Baltimore, MD 21205, USA

**Keywords:** spinal meningioma, cord compression, functional outcome, McCormick scale, magnetic resonance imaging, tumor volume

## Abstract

**Simple Summary:**

Patients affected by spinal meningioma globally exhibit an excellent postoperative functional outcome despite tumor size and severity of spinal cord compression. This work aims to analyze the impact of tumor size and other radiological findings on clinical functional preoperative status and postoperative outcome. In this multicentric retrospective study involving 90 adult patients, we found that spinal cord and tumor occupancy as well as cord compression and tumor volume are correlated with low preoperative functional status. Higher tumor occupancy and tumor-canal volume ratio corresponded to lower postoperative neurological recovery. Cord re-expansion did not show any correlation with postoperative outcome, while preoperative signs of cord myelopathy were predictors of worse postoperative outcome. These findings suggest that these radiological features should be taken into consideration during preoperative counselling.

**Abstract:**

Background: Radiological parameters predicting the postoperative neurological outcome after resection of a spinal meningioma (SM) are poorly studied, with controversial results. Methods: Observational multicenter cohort (2011–2018) of adult patients undergoing surgery for resection of SM. Tumor-canal volume ratio (TCR), the areas related to the cord and tumor occupancy at maximum compression, the presence of dural tail, calcifications, signs of myelopathy, and postoperative cord expansion were compared with the modified McCormick scale (mMCS) preoperative and at follow-up. Results: In the cohort (*n* = 90 patients), cord and tumor occupancy as well as cord compression and tumor volume showed a correlation with preoperative mMCS (*p* < 0.05, R −0.23; *p* < 0.001, R 0.35; *p* < 0.005, R −0.29; *p* < 0.001, R 0.42). Cord occupancy had a strong correlation with cord compression (*p* < 0.001, R 0.72). Tumor occupancy and TCR were correlated with relative outcome at follow-up (*p* < 0.005 R 0.3; *p* < 0.005 R 0.29). No correlation was found between cord re-expansion and clinical outcome at follow-up. Finally, a correlation was shown between preoperative signs of cord myelopathy and mMCS (*p* < 0.05 R 0.21) at follow-up. Conclusions: Larger tumors showed lower preoperative functional status and a worse clinical outcome. Moreover, preoperative T2 cord signal changes are correlated with a poorer outcome.

## 1. Introduction

Spinal meningiomas (SMs) are benign and slow-growing lesions accounting for 25% to 46% of intradural extramedullary lesions [1,2,3]. Most of the affected patients are aged 60 to 80 years old, with a predilection for women over men, with a ratio up to 9:1 [1,4,5,6,7,8,9,10]. These lesions are generally circumscribed, provoking symptoms when compressing the spinal cord, and the neuroradiological investigations are often delayed until altered sensibility, gait disturbances, or sphincteric dysfunction become manifest [6,11]. Despite tumor dimensions and severity of cord compression, most of the patients exhibit an excellent postoperative neurological outcome, notwithstanding the older age [9,12]. Several factors have been associated with a poor outcome, such as psammomatous type of meningioma [4] and WHO grade >1 [13,14], invasion of arachnoid/pia mater [14], Simpson grade resection II and III [13,14,15], ventral attachment [8,13,15,16], calcifications [8,16,17], dural tail and T2 cord signal changes [8], duration of symptoms [13,14], poor preoperative functional status [8,13,14], and sphincteric involvement [13]. Few studies have explored the relations between tumor dimensions, cord compression, and functional outcome, thus reporting contrasting results [4,8,9,14,18,19,20], as illustrated in Table 1. The aim of this study was to perform a thorough analysis of tumor dimensions in terms of tumor area and volume, cord compression, radiological findings, and their correlations with clinical presentation and outcome.

## 2. Materials and Methods

### 2.1. Study Design and Participants

An observational multicentric cohort study conducted at three tertiary referral neurosurgical centers, including patients who underwent SMs resection surgery between January 2011 and December 2018.

Inclusion criteria were: (1) age between 18 and 85 years old, (2) surgery for resection of SM, and (3) available preoperative MRI (T1-weighted with contrast and T2-weighted sequences). Exclusion criteria were: (1) follow-up <12 months after first surgery and (2) lesions involving the craniovertebral junction (according to Setzer et al. [14]), as they represent a different entity [22,23]. All the patients were operated by posterior microsurgical approach, and the resection was classified according to Simpson grading system [24]. The manuscript was written according to the Strengthening the Reporting of Observational Studies in Epidemiology checklist.

### 2.2. Variables and Data Sources

The electronic medical records and MR images were sought to obtain the following data:DemographicsPreoperative functional status according to the modified McCormick Scale (mMCS) [25] and the presence of any neurogenic bladder or bowel dysfunctionTumor location (cervical, thoracic, and lumbar), its attachment (anterior, anterolateral, and posterior), and craniocaudal extension (number of spinal levels)The presence of calcifications, dural tail, foraminal involvement, and T2-weigthed MRI intensity changes (i.e., signal changes) of the spinal cord. The latest was classified according to its extension in: mild (<1 level), moderate (≥1 level but <2 levels), severe (≥2 levels).Tumor volume (cm^3^) manually segmented using Brainlab Elements^®^ on axial T1- weighted images with contrast (slice thickness from 3 to 4 mm) (Figure 1).Canal volume (cm^3^) manually segmented using Brainlab Elements^®^ on axial T2- weighted images (slice thickness from 3 to 4 mm) to assess the volume of the corresponding spinal canal segment (identified by the segment between two axial slices corresponding to the upper and lower limits of the meningioma) (Figure 1).Ratio (%) between tumor and spinal canal volume, identified as tumor canal ratio (TCR) = $\frac{\mathrm{tumor}\;\mathrm{volume}}{spinal\;canal\;volume}\times 100$Areas (mm^2^) related to tumor and cord occupancy at maximum compression level and the degree of spinal cord compression, applying the formula reported by Davies et al. [19] and illustrated in Table 1, with the other previous studies reporting SMs volume analysis. The results are reported as % according to the author. This was calculated using OsiriX Lite^®^ software, based on axial T1-weighted images with contrast and axial T2-weighted images (Figure 2).The degree of postoperative cord expansion (%) at the level of preoperative maximum compression calculated as above based on axial T2-weighted images.T2-weighted images changes of the spinal cord on postoperative MR images.

Preoperative and postoperative radiological imaging was retrospectively analyzed by a neuroradiologist blinded to any clinical data (MA). Measurements were obtained by two authors (VB and AM) also unaware of the clinical status of the patients. When in doubt, measures were discussed and reviewed and solution obtained by consensus with the neuroradiologist (MA).

Intraoperative data: Simpson grade of resection, arachnoidal invasion, use of neuromonitoring, complications;WHO grade;Postoperative complications, complications needing further surgery, recurrence, mMCS at follow-up, length of follow-up;Relative outcome (Δ Outcome) = $\frac{\left(\mathrm{preoperative}\;\mathrm{mMCS}-\mathrm{postoperative}\;\mathrm{mMCS}\right)}{\;\mathrm{preoperative}\;\mathrm{mMCS}}\times 100$

### 2.3. Statistical Analyses

All statistical data were analyzed using TIBCO Statistica^®^ 13.3.0. Results are presented as median and range or as number of cases and percentage. Spearman Rank–Order Correlations were applied.

The following correlations were studied:tumor and cord dimensions with preoperative mMCS and duration of symptoms;tumor and cord dimensions with postoperative mMCS and relative outcome;tumor and cord dimensions with cord re-expansion;cord re-expansion with postoperative mMCS and relative outcome; andradiological features with postoperative mMCS and relative outcome.

Statistical significance was set at *p* < 0.05.

This study was conducted in accordance with the ethical standards of the institutional research committees, ethic committees (Ethics Committee CCER Genève protocol code 2018-02254), and with the 1964 Declaration of Helsinki plus later amendments.

## 3. Results

### 3.1. Patient and Tumor Characteristics

A total of 125 patients were identified, among which 90 patients met the inclusion criteria; each center participated with 42%, 37%, 21% of cases, respectively (16.7% men, median age 67 years, range 25–85), with a male-to-female ratio of 1:5. The median duration of symptoms was 9 months, ranging from 9–60. Functional state at diagnosis was good (mMCS I and II) for 53 patients (58.9%), 14 patients (15.5%) had a poor functional state (mMCS IV and V), and only 13 (14.4%) of patients presented neurogenic bowel/bladder disturbances. Most of the tumors were in the thoracic region (81.1%), and only 2.2% of cases had an extension of more than two levels. Lateral attachment was the most frequent, reported in 37.8% of cases, whereas antero-lateral and posterior location accounted for 24.4% of cases; foraminal involvement was present in 22.2% of patients. The images displayed calcifications and dural tail in 41.1% and 54.4% of the cases, respectively. Radiological signs of myelopathy were present at diagnosis in 61 patients (67.8%) and were classified as mild, moderate, and severe (58.9% vs. 7.8% vs. 1.1%). At maximum compression level, the median tumor occupancy was 73.10% (range 17.20–95.40%), the median cord occupancy 19.40% (range 0.20–56.80%), and the median cord compression 69.75% (range 0.55–169.30%). The median TCR was 56.0% (range 9.0–86.40%). All preoperative data are shown in Table 2.

### 3.2. Intraoperative Findings

Gross total resection (Simpson grade I and II) was achieved in 86.7% of the cases, surgeries were performed without intraoperative complications in almost all the patients (91.1%), and intraoperative neuromonitoring was adopted in nearly half of the cases (46.7%). In eight patients (8.9%), an intraoperative complication was reported. The pathological finding diagnosed a typical meningioma WHO grade I in 94.5% of the tumors. All intraoperative data are shown in Table 3.

### 3.3. Postoperative Outcomes

Clinical follow-up was performed for all the patients (median follow-up duration 17 months, range 3–75), while clinical and radiological follow-up was available for 48 patients (median follow-up duration 19 months, range 3–75). Postoperative complications occurred in 11 patients (12.2%), requiring surgery in 63.6% of them (*n* = 7 out of 11). Good outcome (mMCS grade I and II) was achieved in 83.4% of the patients at last follow-up, with a median relative outcome of 33.3% (range −100–75%). A total of 8.9% of the patients had worsened at last follow-up. Postoperative radiological findings showed mild spinal cord myelopathy in half of the patients (*n* = 24), moderate myelopathy in the 14.6% of patients (*n* = 7), and severe myelopathy in 2.1% of the patients (*n* = 1). A total of 33.3% of the patients (*n* = 16) showed no signs of myelopathy. We observed a median cord expansion of 87.40% (range 57.0–187.60%). All postoperative clinical and radiological data are shown in Table 4.

### 3.4. Influence of Cord Compression and Radiological Features on Preoperative Functional Status and Outcome

1. Tumor and cord dimensions with preoperative mMCS and length of symptoms

Patients’ preoperative functional status showed a very low statistical correlation with cord occupancy (*p* < 0.05, R = −0.23), tumor occupancy (*p* < 0.001, R = 0.35), and cord compression (*p* < 0.005, R = −0.29), whereas its correlation with tumor volume was mild (*p* < 0.001, R = 0.42). Furthermore, we compared cord occupancy with cord compression, highlighting a high correlation (*p* < 0.001, R = 0.72). No correlation was found between preoperative measurements and duration of symptoms.

2. Tumor and cord dimensions with postoperative mMCS and relative outcome

No correlation was found between preoperative measurements and postoperative functional outcome. Nevertheless, a very low correlation between tumor occupancy and TCR with postoperative relative outcome was disclosed (*p* < 0.005, R = 0.3; *p* < 0.005, R = 0.29).

3. Tumor and cord dimensions with cord re-expansion

A correlation between cord and tumor occupancy with postoperative cord re-expansion was found: mild and very low, respectively (*p* < 0.05, R = 0.39; *p* < 0.05, R = −0.31).

4. Cord re-expansion with postoperative mMCS and relative outcome

We found no correlation between cord re-expansion and postoperative outcome measures.

5. Radiological features with postoperative mMCS and relative outcome

A very low correlation between preoperative T2 cord-signal changes and postoperative mMCS was displayed (*p* < 0.05, R = 0.21).

## 4. Discussion

In this multicentric retrospective study involving 90 adult patients operated on for resection of a spinal meningioma, we showed that: (1) high cord and tumor occupancy as well as cord compression and tumor volume are correlated with low preoperative functional status, (2) high cord occupancy presented a high correlation with cord compression, (3) higher tumor occupancy and TCR corresponded to lower postoperative neurological recovery (represented by relative outcome), and (4) cord and tumor occupancy were correlated to postoperative cord re-expansion even if (5) cord re-expansion did not show any correlation with postoperative outcome, while (6) preoperative signs of cord myelopathy were predictors of worse postoperative outcome. In this paper, we summarized previous literature evaluating the tumor dimensions of SMs and the relations with clinical presentation and outcome, highlighting contrasting results among the series. All these studies present different methods of image analysis, which makes difficult to compare their results. With this study, we report the largest multicentric series of SMs with detailed volumetric and non-volumetric radiological analysis.

We highlighted a correlation between preoperative cord and tumor dimensions with preoperative mMCS: stronger for the tumor volume rather than the areas. This finding may be explained by the fact that the compressive process takes place on multiple levels and not only at the point of maximum compression. Nevertheless, the influence of tumor occupation ratio and cord flattening on clinical presentation was reported by Yamaguchi et al., who defined an occupation ratio of 64% as a threshold for the development of motor weakness [20], whereas volume of the tumor influenced only preoperative back pain in the series by Schwake et al. [9]. The measurement performed in the analysis of tumor compression varies among authors, including simple measures, such as the maximum diameter of the tumor [4,18], the percentage of tumor occupancy in the spinal canal at maximum compression [8,14,21], comprehensive cord and tumor areas at maximum compression [19,20], and the calculation of the volume of the tumor applying an estimation formula [9,20]. In order to obtain an extensive and detailed analysis, we applied the measurement proposed by Davies et al. [19], finding high correlation between the cord cross-sectional area and the cord occupancy at maximum compression; thus, one may choose one of them to measure the cord area. Moreover, we used a software applied for neuronavigation and imaging processing to obtain the volume of spinal canal and tumor to overcome the approximation of spheroid volume formula, as already demonstrated for other tumors [26,27].

Concerning the influence of tumor dimensions on outcome, Arima et al. [18] and Maiti et al. [8] found that larger tumors were associated with poorer outcome; similarly, we described a correlation, although very low, with tumor measurements and relative functional outcome. Moreover, postoperative cord expansion at the level of preoperative maximum compression was found to be correlated with preoperative cord compression and tumor occupancy: the lower the cord compression, the higher the postoperative expansion. This could be explained with the fact that chronic spinal compression causes an irreversible neuronal degeneration due to mechanisms linked to ischemic necrosis, the loss of neuronal cells, and the demyelination of the spinal columns [28]. The possible gliosis caused by ischemia can lead to incomplete recovery despite the removal of the tumor [29]. We found no studies that specifically investigated the pathophysiology of compression damage by SMs, but according to the mechanism, it is possibly due to an obstruction of the epidural venous plexus, with consequent vasogenic edema, which results in ischemic damage and subsequent cytotoxic edema [30,31] and an inflammatory reaction caused by the tumor on the arachnoid [32]. In addition, the presence of the neoplasm interferes with the normal movements of the spinal cord, which occur during the movements of the spinal column, and this mechanism contributes to spinal damage [29]. Nevertheless, in our series, cord re-expansion had no influence on outcome, as previously described by Davies et al. [19].

Dural attachment seems to be more frequently localized in lateral [8,17,20] and anterolateral [6,14] region, but only a few studies analyzed specific MR features of the tumor and the compressed cord. Although dural tail was found in up to 43% of patients [8,9,18], correlating with a poor prognosis according to Maiti et al. [8], it was described in more than half of our patients, thus without statistical significance. Moreover, calcifications were highlighted in a minority of patients as an intraoperative finding related to a poorer outcome [5,13,17], whereas our series of preoperative MRI revealed the presence of calcifications in 41.1% of the cases without correlation with functional outcome. Preoperative T2-weighted images alterations of the spinal cord, suggesting myelopathy, were displayed in 39.1% of the cases described by Arima et al. [18] and in 26.3% of the patients reviewed by Maiti et al. [8], who found a correlation with a poor prognosis. In our series, 67.8% of the patients presented from mild to severe signs of myelopathy, showing a correlation, though very poor, with mMCS at follow-up.

Our results seem consistent with previous studies. In fact, most of the SMs were diagnosed at the thoracic level [1,5,6,14,16,33,34] in patients between the VI and VIII decades. Moreover, probably due to hormonal factors, women are more frequently affected than men [1,5,6,12,16,17,32,35]. The median duration of symptoms in the present series was 12.23 months ± 12.12 and correlates well with the time reported in other papers, confirming that a delayed diagnosis of SM is still frequent [5,6,12,16,32,34].

Gross total resection (Simpson I e II) of SMs is globally achieved in more than 90% of the cases previously reported [7,8,12,13,14,16,17,18], slightly higher than in our series. This finding could be related to the more complicated pathology and/or patients treated in referral centers. Pathological analysis revealed a meningioma WHO grade I in 87.6 to 100% of the cases [6,7,8,14,17,33], and our finding of 94.5% is in line with the previous literature. Intraoperative neuromonitoring, adopted in nearly half of our cases, may be useful in SM resection, but it is not mandatory to achieve a safe resection. In fact, it was used in a minority of recent reports without correlation with outcome [8,9,17].

Perioperative complications were observed in 12.2% of all cases, which generally matches findings from previous studies, reporting a surgical morbidity between 1.5 to 13%. Among them, CSF leakage and wound complication were the most frequently encountered [7,9,13,14,17]. Satisfactory outcome (mMCS grade I and II) was achieved in 83.4% of the patients at last follow-up (median duration 17 months, range 3–75). In detail, 8.9% of our patients functionally worsened, findings which are in line with previous studies reporting from 2 to 10% of clinical deterioration in their cases [1,13,14,16,17,34].

While several studies identified factors correlated with a poorer outcome after SM removal, only a few authors thoroughly analyzed the correlation between tumor dimensions and cord compression on clinical presentation and outcome [9,19,20]. Furthermore, the comparison among them is difficult because of the use of different functional scale and different measurement performed (Table 1). The functional scales adopted for the analysis in previous literature were the McCormick Scale [14], the mMCS [8,9,18], and the Nurick scale [19,21], whereas Yamaguchi et al. studied the different types of symptoms without referring to a functional classification scale [20]. We decided to apply the mMCS in our series of patients because it is considered the standard outcome tool in case of patients affected by intradural spinal tumors [25]. Moreover, it is easy to calculate, and it assesses the global functional impairment specifically regarding neurological deficits, gait disturbances, and patient’s autonomy [25,36].

### Limitations

The interpretation of the present results should be considered under some limitations. The retrospective design of the study and the exploratory design of statistical analyses with inherent selection and treatment biases (for example, SM that were selected for clinical-radiological follow-up were not included) may limit the generalizability of the results. The potential bias induced by volumetric estimation or empiric measurements were reduced by using a dedicated software and double checking by the authors (VB and AM).

## 5. Conclusions

According to our findings, patients with larger spinal meningiomas and higher spinal cord compression may present a lower preoperative functional status and may be prone to a worse clinical outcome. Cord re-expansion at follow-up does not represent a predictive factor for neurological recovery after surgery for spinal meningiomas in our series. Moreover, preoperative T2 cord-signal changes are correlated with a poorer outcome. Nevertheless, our results are not sufficient to support aggressive surgical indication in case of asymptomatic patients (e.g., coincidental diagnosis). These findings should be taken into consideration and pushed forward in further studies to obtain strong predictive factors that could guide preoperative counselling.

## Figures and Tables

**Figure 1 cancers-13-04183-f001:**
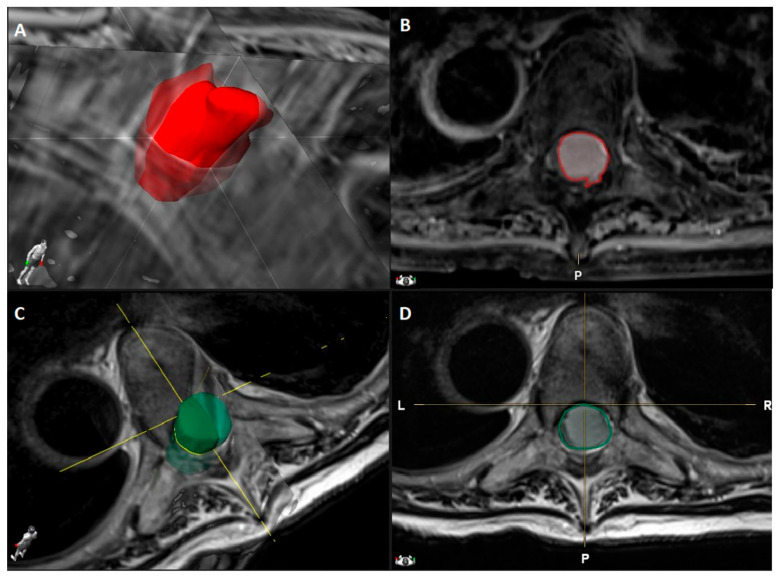
Dorsal Th5-Th6 WHO grade I meningioma with anterior dural attachment: (**A**) 3D tumor volume rendering (2.6 cm^3^) obtained by (**B**) tumor segmentation on axial T1-weighted MRI images after contrast administration using Brainlab Elements^®^; (**C**) Spinal canal volume rendering (3.5 cm^3^) obtained by (**D**) spinal canal segmentation on axial T2-weighted MRI images using Brainlab Elements^®^.

**Figure 2 cancers-13-04183-f002:**
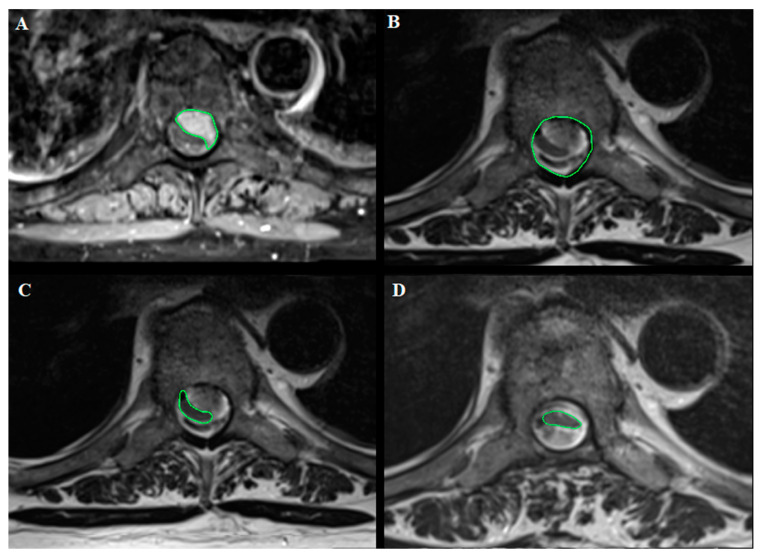
Dorsal Th7 WHO grade I meningioma with anterior dural attachment: (**A**) Tumor area (111 mm^2^) measured at maximal compression on axial T1-weighted images after contrast administration; (**B**) spinal canal area (210 mm^2^) measured at maximal compression on axial T2-weighted images; (**C**) spinal cord area (39 mm^2^) measured at maximal compression on axial T2-weighted images; (**D**) postoperative spinal cord area (42 mm^2^) measured at the level of preoperative maximal compression on axial T2-weighted images. The measurements were made using OsiriX Lite^®^.

**Table 1 cancers-13-04183-t001:** Review of the previous literature. Parameters and formulas are reported as indicated in materials and method section of the respectively papers.

Reference	No. Pts	Dimension Parameters	Measurement	Correlation Found
Schaller et al., 2005 [4]	33	Tumor larger diameter	Manual measurement on CT or MR images (mm)	None
Setzer et al., 2007 [14]	80	Tumor extension	Occupancy > Or <50% of the spinal canal	None
Ahn et al., 2009 [21]	10	Percentage of tumor occupying the intradural space	((the transverse diameter of the tumor mass + the longitudinal diameter of the tumor mass)/(the transverse diameter of the intradural space + the longitudinal diameter of the intradural space)) × 100 on the axial image showing the maximum size	None
Arima et al., 2014 [18]	23	Tumor larger diameter	Manual measurement on MR images. Tumor larger diameter > than 2/3 of spinal canal diameter	Larger tumor were associated with poorer outcome
Maiti et al., 2016 [8]	38	Tumor larger diameter	Manual measurement on MR images. Tumor occupying >75% of spinal canal in AP and transverse direction	Larger tumor were associated with poorer outcome
Davies et al., 2016 [19]	31	1. Cord area (%) at maximum compression = cross sectional area 2. Tumor occupancy (%) at maximum compression 3. Cord occupancy (%) at maximum compression	1. Cord area at maximum compression (%) Area of cord at maximum compression × 100 (Area of cord above + area of cord below)/2 2. Area of tumour at maximum compression (%) Area of tumor at maximum compression × 100 Area of canal at maximum compression 3. Area of cord at maximum compression (%) Area of cord at maximum compression × 100 Area of canal at maximum compression	None
Schwake et al., 2018 [9]	88	1. Spinal cord compression 2. Tumor volume	1. Manual measurement on T2-weighted MR images. 2. Analyzed on T1-weighted Gadolinium-enhanced images and estimated using the formula for a spheroid V = 4/3 × *p* × r1 × r2 × r3. In this formula, r is the radius, which is 50% of the tumor diameter at the site of its largest extension in axial, coronal, and sagittal planes	An increasing volume was correlated to preoperative low back pain
Yamaguchi et al., 2020 [20]	53	1. Cross-sectional area (mm^2^) of the spinal canal and that of the tumor 2. Occupation ratio of the tumor in the spinal canal (%) 3. Spinal cord flattening ratio (%) 4. Estimated tumor volume	1. Cross-sectional area (mm^2^) of the spinal canal and that of the tumor measured on the axial T1-weighted, contrast-enhanced MR image that showed the highest degree of spinal cord compression 2. Occupation ratio (%) = (tumor area (mm^2^)/spinal canal area (mm^2^)) × 100 3. 100 − (area of spinal cord at maximum compression/(area of cord above + area of cord below)/2) × 100 4. Volume (mm^3^) = (π × height × width × depth)/6	1. Occupation ratio and spinal cord flattening ratios were significantly larger in the presence of motor weakness or sensory disturbances. 2. Occupation ratio is an independent contributing factor to the presentation of motor weakness and pain. 3. An occupation ratio of approximatively 64% could be used as a threshold value of tumor grow to cause motor weakness.

**Table 2 cancers-13-04183-t002:** Preoperative clinical and radiological features of the patient cohort.

**No of patients**	**90 (100%)**
**Clinical Data**	
Sex	
Female (F)	75 (83.3%)
Male (M)	15 (16.7%)
Ratio F:M	5:1
Age, years, median (range)	67 (range 25–85)
NF 2	2 (2.2%)
Duration of symptoms, months, median (range)	9 (range 0–60)
Modified McCormick scale	
I	27 (30%)
II	26 (28.9%)
III	23 (25.6%)
IV	13 (14.4)
V	1 (1.1%)
Sphincteric dysfunction	13 (14.4%)
**Radiological parameters**	
Tumor location	
Cervical	13 (14.5%)
Cervico-dorsal	2 (2.2%)
Dorsal	73 (81.1%)
Dorso-lumbar	1 (1.1%)
Lumbar	1 (1.1%)
Attachment of the tumor	
Anterior	7 (7.8%)
Antero-lateral	22 (24.4%)
Lateral	34 (37.8%)
Postero-lateral	5 (5.6%)
Posterior	22 (24.4%)
Foraminal invasion	20 (22.2%)
Calcification	37 (41.1%)
Dural tail	49 (54.4%)
MRI T2-weighted images changes of the spinal cord	
None	29 (32.2%)
Mild	53 (58.9%)
Moderate	7 (7.8%)
Severe	1 (1.1%)
Single lesion	87 (96.7%)
Craniocaudal tumor extension	
1 level	55 (61%)
2 level	33 (36.7%)
3 level	2 (2.2%)
Analysis of tumor and cord dimensions, median (range)	
Tumor occupancy at maximum compression	73.10% (17.20–95.40%)
Cord occupancy at maximum compression	19.40% (0.20–56.80%)
Cord compression of the estimated original value	69.75% (0.55–169.30%)
Tumor Canal Ratio (TCR)	56.0% (9.00–86.40%)

MRI, magnetic resonance imaging; NF, neurofibromatosis.

**Table 3 cancers-13-04183-t003:** Intraoperative findings of the patient cohort.

No. of Patients	90 (100%)
Extent of resection	
Simpson I	8 (8.9%)
Simpson II	70 (77.8%)
Simpson III	10 (11.1%)
Simpson IV	2 (2.2%)
Intraoperative complications	8 (8.9%)
Nerve root lesion	6 (6.7%)
Bleeding with hypotension	1 (1.1%)
Bradycardia with hypotension	1 (1.1%)
Use of intraoperative neuromonitoring	45 (46.7%)
Arachnoidal invasion	1 (1.1%)
Histopathological grading	
WHO I	85 (94.5%)
WHO II	4 (4.4%)
WHO III	1 (1.1%)

WHO, World Health Organization.

**Table 4 cancers-13-04183-t004:** Postoperative clinical and radiological features of the patient cohort.

**Clinical data**	
No. of patients	90 (100%)
Complications	11 (12.2%)
None	79 (87.8%)
CSF leak	6 (6.7%)
Hemorrhage	5 (5.6%)
Wound dehiscence	1 (1.1%9
Surgery for complications (on 11 patients)	7 (63.6%)
Recurrence	2 (2.2%)
Modified McCormick scale	
I	61 (67.8%)
II	14 (15.6%)
III	12 (13.3%)
IV	3 (3.3%)
V	-
Δ Outcome, median (range)	33.3% (−100–75%)
Improved	53 (58.9%)
Stable	29 (32.2%)
Worsened	8 (8.9%)
Length of follow-up, months, median (range)	17 (3–75)
**Radiological parameters**	
No. of patients	48
MRI T2-weighted images changes of the spinal cord	
None	16 (33.3%)
Mild	24 (50%)
Moderate	7 (14.6%)
Severe	1 (2.1%)
Spinal cord re-expansion, median (range)	87.40% (57.0–187.60%)

CSF, cerebrospinal fluid; MRI, magnetic resonance imaging; SD, standard deviation.

## Data Availability

All data are available in the text.

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
