# Peer review of "Spinal Meningiomas: Influence of Cord Compression and Radiological Features on Preoperative Functional Status and Outcome"

_cancers, 2021, doi:10.3390/cancers13164183_

Round 1

Reviewer 1 Report

Dear authors,

large multi-center study on spinal meningeomas. High case number. Meticulous MRI analysis. (Expectable) findings comparable to other larger size studies. Maybe the authors could add a short paragraph on how the results may directly (pragmatic approach) affect the consultation of patients?! Do results speak in favour of a rather "proactive" resection in asymptomatic cases (e.g. coincidental findings) before size increase and MRI changes occur?

Author Response

Dear Reviewer,

thank you for your positive comment. Unfortunately, the weak correlations found in our analysis are not sufficient to promote a “proactive” resection in case of asymptomatic patient. As suggested, we add a brief paragraph about this issue (conclusion section). 

Reviewer 2 Report

The authors present their series on 90 patients with spinal meningiomas who underwent resection and outcomes based on imaging findings. They use the opportunity to review the literature for comparison. They find a correlation between pre-op functionality and cord occupancy, tumor occupancy, cord compression, and tumor volume (albeit low). A low correlation of TCR and tumor occupancy with the relative outcome. Low correlation between cord and tumor occupancy and re-expansion and low correlation between preop imaging signs of myelopathy and mMCS.

We applaud the authors for a very thorough review on the topic of spinal meningiomas. We believe it is fitting for this special issue of Cancers. We however have a few questions/concerns to mention. All the correlations found by the authors are very weak at best and we are concerned they do not have enough power to profoundly change pre-operative counseling. The weak correlation between variables begs the question if the issue is due to sample size. While the authors make a claim that this topic has not been extensively reported on (line 62-64), table one shows robust literature for a relatively rare pathology. We offer the authors to consider performing a meta-analysis.

The authors have pathology results but did not analyze outcomes based on this. Is there a reason for this? Peri-op complications and functional decline were slightly higher than anticipated in this series and gross total resection rates lower (though the manuscript has references in which this falls within the range reported in the literature). Some discussion on this may be warranted (centers represented in this series are referral centers so more complicated patients or pathology? etc)

Author Response

Dear Reviewer,

thank you for your positive comment. The sample size was appropriated for the statistical analysis. Prof. Mario Ermani, one of the authors who performed statistical analysis for this study, is a neurologist and the biostatistician of the Department of Neuroscience, Università degli Studi di Padova.

Concerning the literature, only three studies analyzed, with different degree of depth, areas and volumes (Davies et al, Schwake et al, Yamaguchi et al). We took the opportunity to carry a comprehensive analysis of both measurements. Due to the heterogeneity of the parameters used (as highlighted in Tab. 1) we could not perform a meta-analysis but only a descriptive review of the pertinent literature.

In this paper we focused our analysis only on cord compression and radiological features. The correlation with outcome and the other variable, such as histological subtypes, are going to be further analyzed and eventually presented in a separated paper.

Peri-operative complications and functional decline are in line with literature, thus towards the highest values (i.e. complications 12.2% vs 1.5-13% reported in literature; functional decline 8.9% vs 2-10% reported in literature) whereas the rate of subtotal/debulking (identified as Simpson grade II and IV) where slightly higher. These findings could be effectively related to the more complicated pathology and/or patients treated in referral centers. We add a brief comment on this issue on page 11 line 356-359.

Reviewer 3 Report

The study investigates if dimensional analysis of the spinal meningiomas in radiology images correlates with the post-surgery clinical outcome. The dimensions are measured using a commercial software retrospectively on a population of 90 patients from three institutes. The aim of the study is clear and a fair review of the literature is included however the novelty of the study is not well-motivated compared to the state of the art and literature. The paper mentions two major conclusions: larger tumors may be, and cord signal changes are, correlated to a poorer clinical outcome.

Overall comments:

  • There should be a section in the material and methods explaining the dimensional calculations for the area and volume and the units assigned to them, as well for all other metrics used. The results present the values in percentage only, making it difficult to assess the original values.
  • The quality of presentation of the material and methods and results should in general be improved to make the paper more readable.
  • Statistical analysis: Several groups of data do not seem to have a normal distribution, therefore mean+/- SD is not the right measure, median and range should be used instead/ besides the mean +/- SD. In addition to the correlation measure, the statistical significance tests should be deployed to investigate the hypothesis. A statistician can be consulted in the choice of statistical methods.
  • ‘Signal changes’ in T2 images is mentioned as one parameter leading to the conclusion of a poorer outcome. The assessment of ‘signal changes’ should be explained in the material and methods and added to the aim of the study and any other relevant section.
  • The conclusion needs to be improved as it is currently loose and not well reflecting on the results and the aim, e.g., no conclusion is given on the effect of cord compression
  • The article needs proofreading.

Detailed comments:

Page 2, The below two statements are contradictory, and the first one demotivates the reason for performing the study as the result is already known.

Lines 55-57: “Despite tumor dimensions and severity of cord compression, most of the patients exhibit an excellent postoperative neurological outcome, notwithstanding the older age”

Lines 62-64: “Few studies have explored the relations between tumor dimensions, cord compression and functional outcome, thus reporting contrasting results”

Page 2, Participants: Mention how many patients in total and in each sub group were included.

Page 2. Study design should include more information about the study, parts of which are currently under participants. Please expand on this section.

Lines 85-86: Please give a general view on the MR data by mentioning all the MR data modalities used. If available please also mention the range of the slice thickness of the MR images as that can affect the precision of volume calculations.

Page 2, variables and data sources. Please consider a different structure for presenting the information currently in the bullet point list. Perhaps a table will be a better way of presenting the data vs variables.

Line 139: please mention if an ethical approval was obtained or not needed (even if repeated at the end of the manuscript).

 Page 4, lines 144-145: the information regarding patients and any other information that is not ‘a result’ should be included in the material and methods.

Line 160: How was the gross total resection assessed/calculated? This parameter in principle should be a post operative parameter if not based on the intraoperative radiology.

Lines 179-180: The sentence is incomplete or needs punctuation.

Line 184-201: Please add the = sign in front of the R value or at least keep the expressions consistent

Are some of the R-values negative? It should then be considered that the correlation is inverse.

Lines 275: Should this sentence be negative?

Line 299: Please clarify how the statement in line 298 confirms findings of 94.5%

Tables and Figures:

All the acronyms in the tables should be explained in the table caption or written out in the table, e.g. Table 1: NF, V, F:M

Please check this point for all the tables and figures

Proofread text in Table 1. Some of the text in the table show text similarity to the source.

Figure 1. B) mentions tumor volume measurement, however, the image shows the circumference of the tumor cross section and not the volume. Please add the value of the volume (in cl or cubic mm or any other unit) measured in the illustrated images.

Figure 2. The text in the images is not readable and thus does not provide any information. These should be omitted. Any necessary information can be added separately on the image. The figure caption repeats some information for each sub-image part making it cumbersome to find the important information. Please make the caption concise, e.g. if all the calculations were performed in OsiriX Lite, that can be mentioned once at the start or end of the caption.

D1, D5-7 should be either explained in the text or if not giving any useful information, can be omitted.

The presentation could benefit from adding graphs for illustrating the correlation of tumor dimensions with other quantitative measures, at least where the results are of significant importance for the conclusion.

Text similarity:

The below show a high similarity with the text in the original document:

 Table 1.

  • Third row (Ahn et al.), Column measurement
  • Sixth row (Davies et al.), Column measurement
  • Eighth row (Yamaguchi et al.), column measurement, point 1

Line 320: “global functional impairment in terms of neurological function and walking ability”

Author Response

The study investigates if dimensional analysis of the spinal meningiomas in radiology images correlates with the post-surgery clinical outcome. The dimensions are measured using a commercial software retrospectively on a population of 90 patients from three institutes. The aim of the study is clear and a fair review of the literature is included however the novelty of the study is not well-motivated compared to the state of the art and literature. The paper mentions two major conclusions: larger tumors may be, and cord signal changes are, correlated to a poorer clinical outcome.

Dear Reviewer, thank you for such a careful and thorough reading of this manuscript and for the suggestions that led us to a critical review of our work.

Concerning the novelty added by the study we have discussed it in page 10 lines 280-285.

Reviewer’s Overall comments:

There should be a section in the material and methods explaining the dimensional calculations for the area and volume and the units assigned to them, as well for all other metrics used. The results present the values in percentage only, making it difficult to assess the original values.

The quality of presentation of the material and methods and results should in general be improved to make the paper more readable.

We added the units assigned for areas and volumes. Concerning the dimensional calculations used we better explained the TCR whereas, for the calculations related to the areas we preferred to recall the formula reported by Davies et and reported in detail in Tab.1.

The material and methods and the results section are presented schematically and systematically in order to better organize all the informations. The choice to present variables and data sources in a bullet point list was made to avoid to many tables, which could undermine a fluent reading.

Statistical analysis: Several groups of data do not seem to have a normal distribution, therefore mean+/- SD is not the right measure, median and range should be used instead/ besides the mean +/- SD. In addition to the correlation measure, the statistical significance tests should be deployed to investigate the hypothesis. A statistician can be consulted in the choice of statistical methods.

We change mean values with median values as suggested. Prof. Mario Ermani, one of the authors, is a neurologist and the biostatistician of the Department of Neuroscience, Università degli Studi di Padova (https://elearning.unipd.it/dns/blocks/course_managers/manager.php?id=1846&b=36&lang=en).

‘Signal changes’ in T2 images is mentioned as one parameter leading to the conclusion of a poorer outcome. The assessment of ‘signal changes’ should be explained in the material and methods and added to the aim of the study and any other relevant section.

T2 signal changes are explained in the appropriated section, page 2 lines 92-95. Since it is an MRI finding it is considered with the other radiological parameters.

The conclusion needs to be improved as it is currently loose and not well reflecting on the results and the aim, e.g., no conclusion is given on the effect of cord compression

We revised and strengthened the conclusion section to better fit with the aim of the results of the study basing on reviewers’ comments.

The article needs proofreading.

A native English speaker revised the article.

Reviewer’s Detailed comments:

Page 2, The below two statements are contradictory, and the first one demotivates the reason for performing the study as the result is already known.

Lines 55-57: “Despite tumor dimensions and severity of cord compression, most of the patients exhibit an excellent postoperative neurological outcome, notwithstanding the older age”

Lines 62-64: “Few studies have explored the relations between tumor dimensions, cord compression and functional outcome, thus reporting contrasting results”

Page 2 line 55-57 and 62-64: These statements are not contradictory. The majority of the patient affected by a spinal meningioma globally present a good neurological outcome but some other not. It is important to investigate the reason leading to a worse outcome in this part of patient. Moreover, as highlighted in the introduction and in Tab. 1, the studies that analyzed the influence of tumor dimension on outcome considered different parameters, are difficult to compare and they reach different conclusions.

Page 2, Participants: Mention how many patients in total and in each sub group were included.

Page 2. Study design should include more information about the study, parts of which are currently under participants. Please expand on this section.

Page 2: we added the information requested.

Lines 85-86: Please give a general view on the MR data by mentioning all the MR data modalities used. If available please also mention the range of the slice thickness of the MR images as that can affect the precision of volume calculations.

Line 85-86: We added the slide thickness. The bias related to the slice thickness is already reported in the “study limitations section”.

Page 2, variables and data sources. Please consider a different structure for presenting the information currently in the bullet point list. Perhaps a table will be a better way of presenting the data vs variables.

Page 2, variables and data sources: answered above.

Line 139: please mention if an ethical approval was obtained or not needed (even if repeated at the end of the manuscript).

Line 139: we added the required information.

 Page 4, lines 144-145: the information regarding patients and any other information that is not ‘a result’ should be included in the material and methods.

Page 4, lines 144-145: since the information presented derives from the application of inclusion and exclusion criteria the authors considered them as part of the result section.

Line 160: How was the gross total resection assessed/calculated? This parameter in principle should be a post operative parameter if not based on the intraoperative radiology.

Line 160: we applied the Simpson grading (intraoperative evaluation of the extend of resection) to assess the extent of resection (https://doi.org/10.1136/jnnp.20.1.22).

Lines 179-180: The sentence is incomplete or needs punctuation.

Lines 179-180: we added punctuation.

Line 184-201: Please add the = sign in front of the R value or at least keep the expressions consistent

Line 184-201: we added the = sign. Some of the R-values are negative because the correlation is inverse.

Are some of the R-values negative? It should then be considered that the correlation is inverse.

Lines 275: Should this sentence be negative?

Lines 275: we corrected the grammar.

Line 299: Please clarify how the statement in line 298 confirms findings of 94.5%

Line 299: we corrected the sentence.

Tables and Figures:

All the acronyms in the tables should be explained in the table caption or written out in the table, e.g. Table 1: NF, V, F:M

Please check this point for all the tables and figures

Tables and Figures: we added all the acronyms used in the figures and tables.

Figure 1. B) mentions tumor volume measurement, however, the image shows the circumference of the tumor cross section and not the volume. Please add the value of the volume (in cl or cubic mm or any other unit) measured in the illustrated images.

Figure 1. B) In Box B and D are presented the areas needed for the manual segmentation process required to obtain the volume using Brainlab Elements®

Figure 2. The text in the images is not readable and thus does not provide any information. These should be omitted. Any necessary information can be added separately on the image. The figure caption repeats some information for each sub-image part making it cumbersome to find the important information. Please make the caption concise, e.g. if all the calculations were performed in OsiriX Lite, that can be mentioned once at the start or end of the caption

D1, D5-7 should be either explained in the text or if not giving any useful information, can be omitted.

Figure 2. We modified the figures and the caption.

The presentation could benefit from adding graphs for illustrating the correlation of tumor dimensions with other quantitative measures, at least where the results are of significant importance for the conclusion.

The correlations studied are defined in the Materials and Methods section, exposed in the Results and subsequently largely Discussed. Considering that the manuscript already includes 4 tables and 2 figures we decided not to include several graphs, which contain statistical representation of the above cited correlations for the risk of rendering the article too long and less straightforward, especially since in our perspective they would not add any benefit to the comprehension of our article. However, if the Reviewer believes that the addition of these graphs could improve the value of the paper, we could add it as supplementary materials.

Proofread text in Table 1. Some of the text in the table show text similarity to the source.

Text similarity:

The below show a high similarity with the text in the original document:

Table 1.

Third row (Ahn et al.), Column measurement

Sixth row (Davies et al.), Column measurement

Eighth row (Yamaguchi et al.), column measurement, point 1

Concerning the text similarity of Tab. 1, we reported in the explanation above the table that parameters and formulas are reported as indicated in materials and method section of the respectively papers, since it is mandatory to meticulously report them, without interpretation.

Line 320: “global functional impairment in terms of neurological function and walking ability”

We corrected the above sentence.

Round 2

Reviewer 3 Report

The authors have implemented most of the comments alternatively motivated their choices. However, there are still some unclarities with details as below: 

Material and Methods

Explanation of ‘Signal changes’ is added to lines 92-95. The authors by signal changes probably mean the intensity changes in the images which they have evaluated visually.

Lines 179-180: Please revise to add a suitable punctuation mark.

Results

The authors have added the metrics to the material and method and Figures but not to the Tables. Therefore, it is still unclear what dimensions the %s represent.  

Statistical analysis

The authors have only used median for the age and have not added the median to the other values in the table. 

Conclusion

No correlation statistics is clearly presented in the result section to support the below sentence. Please add otherwise clarify how this conclusion is supported by the results as the results in Table 4 do not present any values related with 'correlation'.

Moreover, preoperative T2 cord signal changes are correlated with a poorer outcome."

Tables and Figures:

Figure 1. it is not clear what B is illustrating. Part D illustrates the circumference or areas and not volumes as it is stated in the text. Please revise either the text or illustration. 

Text similarity is corrected or motivated.

Author Response

We would like to thank the reviewer for his complete and accurate reading of the revised version of the manuscript and for the meticulous comments, which further improved the quality of this manuscript. Our responses follow.

Material and Methods

Explanation of ‘Signal changes’ is added to lines 92-95. The authors by signal changes probably mean the intensity changes in the images which they have evaluated visually.

Response: we specified that we refer to “T2-weighted MRI intensity changes of the spinal cord” in the description. As you well understand from the explication, the degree of myelopathy was evaluated visually (by a blinded board-certified neuroradiologist), classifying it in term of extension.

Lines 179-180: Please revise to add a suitable punctuation mark.

Response: we correct the grammar and punctuation.

Results

The authors have added the metrics to the material and method and Figures but not to the Tables. Therefore, it is still unclear what dimensions the %s represent. 

Response: as well described in the Material and Methods section, for the analysis of tumor and spinal cord occupancy with the regard to the spinal canal, we applied the formula reported by Davies et al ref. 19. In their work the data are presented as % and it is described in the table referenced in the text. To provide a definitive clarification of this detail, which could lead to confusion an inexperienced reader, we added a brief clarification on page 3 line 108-109.

Statistical analysis

The authors have only used median for the age and have not added the median to the other values in the table.

Response: we corrected the other values in the tables and in the text changing mean +- SD with median and range as previously suggested.

Conclusion

No correlation statistics is clearly presented in the result section to support the below sentence. Please add otherwise clarify how this conclusion is supported by the results as the results in Table 4 do not present any values related with 'correlation'.

Moreover, preoperative T2 cord signal changes are correlated with a poorer outcome."

Response:  T2 cord signal changes of the spinal cord defined the presence of myelopathy. Nevertheless, we changed myelopathy on line 207 with his radiological definition.

Tables and Figures:

Figure 1. it is not clear what B is illustrating. Part D illustrates the circumference or areas and not volumes as it is stated in the text. Please revise either the text or illustration.

Response: to better clarify we specified in the Material and Methods section that the volumes of the tumor and of the spinal canal were obtained by manual segmentation (Lines 101 and 104). Figure 1 B shows the segmentation we made in axial T1-weighted post-contrast MRI to obtain the volume rendering in figure A. The same process is showed in C and D with spinal canal. We have corrected Figure 1 caption to clarify that process.